🔓 | **Open Peer Review** | Clinical Microbiology | Research Article

# CD86 immunohistochemical staining for the detection of *Talaromyces marneffei* in lesions

Jinling Fang,[1] Huiyuan Chen,[1] Krishna Hamal,[1] Donghua Liu[1]

**ABSTRACT** Talaromycosis is an invasive fungal disease caused by the pathogenic thermodimorphic fungus *Talaromyces marneffei* (TM), which is often overlooked in tropical and subtropical regions of Asia. In view of the diversity of clinical manifestations in patients with TM infection, early diagnosis remains challenging. We assessed the sensitivity and specificity of a novel immunohistochemical staining by performing CD86 immunohistochemical staining on 56 tissue sections from patients with talaromycosis who had fungal culture or metagenomic next-generation sequencing confirmed to exist in clinical specimens, as well as 26 patients with other fungi that had been culture-proven. Hematoxylin and eosin and periodic acid Schiff (PAS) stains were also applied to each specimen. We found that anti-CD86 antibody can label TM pathogens in 38 HIV-negative specimens (38/42) and 14 HIV-positive specimens (14/14); conversely, PAS staining yielded positive results in seven cases of HIV-negative specimens (7/42) and 13 cases of HIV-positive specimens (13/14). Additionally, CD86 immunohistochemical staining was negative in other fungi. Importantly, CD86 immunohistochemical staining significantly outperformed PAS staining in terms of localizing and highlighting TM yeasts, as well as demonstrating the specificity of 100% and a significantly higher sensitivity compared to PAS staining at 92.9% versus 35.7% ($P < 0.05$, McNemar test). Our findings suggest that CD86 immunohistochemical staining has the potential for the rapid diagnosis of talaromycosis.

**IMPORTANCE** Talaromycosis is an opportunistic endemic disease without typical clinical manifestations that has emerged as a fungal disease impacting the survival and mortality of immunocompromised individuals and HIV-positive individuals in endemic regions. Nonetheless, talaromycosis is completely curable if it is accurately diagnosed and treated effectively at an early stage. Rapid pathological diagnosis relies on the unique morphological features of *Talaromyces marneffei* observed under the microscope. This study introduces a novel pathological diagnostic approach, CD86 immunohistochemical staining, to enhance the early detection of TM-infected lesions.

**KEYWORDS** *Talaromyces marneffei*, talaromycosis, lesions, CD86 immunohistochemical staining

Talaromycosis, also known as penicilliosis, is caused by the pathogenic thermodimorphic *Talaromyces marneffei* (TM) and is an underrecognized endemic disease prevalent in tropical and subtropical regions, including China, India, and Southeast Asian countries (1, 2). With the widespread use of immunosuppressants and the rise in transplantation procedures, there have been increasing reports of TM infections in HIV-negative patients with diverse immunocompromised conditions, such as autoimmune diseases, tumors, interferon-γ autoantibodies, and solid organ or bone marrow transplantation (3, 4). Due to the atypical clinical features, clinicians often fail to recognize talaromycosis or misdiagnose it as other diseases, such as tuberculosis,

Address correspondence to Donghua Liu, ldhgxmu@163.com.

Jinling Fang and Huiyuan Chen contributed equally to this article. Author order was determined by drawing straws.

The authors declare no conflict of interest.

See the funding table on p. 9.

resulting in severe infections and sporadic diseases affecting various organs, particularly in HIV-negative individuals (5). Talaromycosis predominantly affects HIV-positive patients, with a 16% prevalence in China, and is ranked after tuberculosis and cryptococcosis or pneumocystis as the third most common HIV-associated opportunistic infection in Thailand and Hong Kong (2, 6, 7). Over the last decade, the average annual incidence of *Talaromyces marneffei* was 19,000, and the average number of deaths was 9,000, and the proportion of deaths attributed to fungal infections was as high as 90% (8). As of mid-2022, 34 countries had reported more than 288,000 cases of talaromycosis (1). The morbidity and mortality of talaromycosis remain high in endemic regions and have become a severe endemic health problem. Furthermore, Narayanasamy et al. have proposed to include talaromycosis in the list of fungal pathogens by the World Health Organization as a neglected tropical disease (9). However, the TM infection is completely curable if it is accurately diagnosed and treated effectively at an early stage. While fungal culture remains the gold standard for diagnosis, its diagnostic efficiency is limited and time-consuming. Blood samples are the primary clinical sample for culture; they are missed in 30% of patients living with HIV (10, 11) and 50% of patients without HIV (12). A clinical cohort study from mainland China illustrated that delayed diagnosis is related to an increase in the death rate from 24 to 50% (13). Therefore, more attention should be paid to early diagnosis, and there is an urgent need for rapid and effective diagnosis.

The CD86 molecule (B7-2), with the CD80 molecule, belongs to the B7 family of immune proteins and is expressed in antigen-presenting cells (APCs), such as dendritic cells, monocytes, macrophages, and B cells. As the costimulatory molecules of CD28 and CTLA4, CD86 and CD80 are functionally distinct, but they cooperate with each other in regulating the activation of T cell immunity and playing a particularly significant role in intercellular identification and signaling of APC-T cells (14, 15). In histopathology, anti-CD86 and anti-CD80 antibodies are commonly used to label APC cells to study the relationship between signal transduction between APC and T cells and disease progression, such as allergic diseases(16).

In the early study, we found that TM yeast cells could be labeled with anti-CD86 antibody in tissue sections of patients infected with TM (17). Similarly, we discovered that TM can bind to CD86 monoclonal antibodies in the lung and liver tissues of mouse models infected with TM. In the vitro culture system, when TM infected macrophages, the CD86 molecule in the supernatant decreased significantly at 72 h, while in the control group, it increased; moreover, we also found that TM organisms had the ability to capture CD86 molecule from macrophages (18, 19). Hence, there is a consideration of using the anti-CD86 antibody to label TM yeast cells in lesion tissues as a novel approach for the early diagnosis of talaromycosis. Here, we aimed to investigate the clinicopathological features of TM-infected tissues and evaluate the diagnostic utility of CD86 immunohistochemical staining for talaromycosis.

## MATERIALS AND METHODS

### Specimens

A total of 82 specimens were collected from the First Affiliated Hospital of Guangxi Medical University (Nanning, China). The 56 biopsies were from patients with TM infection, including 50 cutaneous lesions and 6 lymph nodes; the remaining 26 biopsies were skin lesions from patients with other fungal infections, such as *Candida* (4 cases), *Cryptococcus* (4 cases), dematiaceous fungi (4 cases), *Histoplasma capsulatum* (1 case), *Aspergillus* (3 cases), *Sporothrix* (5 cases), *Trichosporon beigelii* (2 cases), *Mucor* (2 cases), *Trichoderma*, and *Fusarium* spp. (1 case). This study was approved by the hospital's medical ethics committee (KT-004).

### Immunohistochemical method

Eighty-two tissue samples were collected and sliced to a thickness of 2–3µm. A total of 26 biopsies were skin lesions from patients with three groups: experimental, negative

section control, and no primary control. Following dewaxing and hydration, sections were immersed in 3% $H_2O_2$ in deionized water and incubated at room temperature for 10–30 min to eliminate endogenous peroxidase activity. The sections were treated with a 5% bovine serum albumin blocking solution to prevent nonspecific binding. Subsequently, the primary anti-CD86 [1:200 (Abcam plc, Cambridge, UK)] was added, and the sections were incubated at 37°C for 2 h. The sections were then individually soaked in the ready-to-use amplification reagent and the ready-to-use HRP-labeled secondary antibody (Mxb Biotechnologies, Fuzhou, China) for 15 min. The reaction time was monitored by observing the staining situation under the microscope, and it was finished by rinsing with tap water. After staining with hematoxylin, water-soluble sealer drops were evenly distributed across sections. Once solidified, an appropriate amount of neutral resin was added to cover the tissue. The no primary control sections were solely incubated with antibody diluent and lacked anti-CD86 molecule antibodies; the negative section control sections were without primary and secondary antibodies; and the remaining procedures were consistent.

### Periodic acid Schiff staining

The section underwent dewaxing and hydration, followed by immersion in a filtered aqueous amylase solution to cover the tissues for 30 min. The section was then treated with a periodic acid solution for 15 min at room temperature to oxidize the samples, followed by the addition of a Schiff reagent for 3–5 min at 37°C. After the tissues turned pink, they were soaked in distilled water for 10 min. Subsequently, the tissue was dehydrated, rendered transparent, and sealed with neutral resin.

### Hematoxylin and eosin staining

All tissue sections were detected by using a hematoxylin–eosin (H&E) stain kit (Solarbio, Beijing, China).

### Statistical analysis

The sensitivity of CD86 immunohistochemical staining and PAS staining on the same specimens was compared using the McNemar test. $P < 0.05$ represents statistical significance. All data were analyzed using the Statistical Package for Social Sciences (SPSS Inc., Chicago, USA, version 26.0 for Windows).

## RESULTS

The TM-infected specimens were collected from 56 patients (42 cases of HIV-negative and 14 cases of HIV-positive) infected with TM. In this population, the median age was 49 (interquartile range: 15–75) years, and the gender ratio of males and females was 35:21. According to the infected organizational types, it was divided into 50 cutaneous tissues and 6 lymphoid tissues. The location distribution of infected specimens was mainly in upper limbs (16, 28.6%), necks (12, 21.4%), and trunks (11, 19.6%), followed by six (10.7%) in the face, six (10.7%) in the lymph nodes, three in the lower limbs and two skin tissues of unknown location (Table 1).

### Histopathologic characteristics of *Talaromyces marneffei* in HIV-negative biopsy specimens

All TM-infected pathological specimens exhibited varying degrees of inflammatory cell infiltration, including lymphocytes, neutrophils, histiocytes, and plasma cells (Fig. 1). No basophilic granular matter or round-like spore structures were identified in hematoxylin and eosin staining. PAS staining revealed positivity in only seven specimens (16.7%), showing oval fungal yeast organisms within histiocytes. Although the individual fuchsia oval yeast shape was visible at high magnification, the morphology was too vague and indistinct to reveal the unique transverse septum (due to division by fission) of

**TABLE 1** Clinicopathological information of cases infected with *T. marneffei*

| Clinicopathological feature | Count |
| --- | --- |
| Age mean (range) | 49 (15–75) |
| Gender | |
| Male | 35 |
| Female | 21 |
| HIV | |
| Positive | 14 |
| Negative | 42 |
| Initial diagnostic way | |
| Cultured specimens | |
| Skin | 32 |
| Pus | 5 |
| Secretions | 4 |
| Blood | 3 |
| Lymph nodes | 3 |
| Bone marrow | 3 |
| Alveolar lavage fluid | 3 |
| mNGS | 3 |
| Localization of specimens | |
| Face | 6 |
| Neck | 12 |
| Upper limbs | 16 |
| Lower limbs | 3 |
| Trunk | 11 |
| Lymph nodes | 6 |
| Unclear skin | 2 |

the TM organism (Fig. 2A and B). Notably, CD86 immunohistochemical staining showed positivity in 38 specimens (90.5%), displaying the typical sausage-like structure of TM. The cell walls of TM yeast cells were continuously and uniformly stained in bright fuchsia, with the cellular center remaining unstained, creating an empty halo. The distinctive morphology of the sausage-like yeast characterized by an oval structure with a cross-wall (the cell walls of two unseparated yeast cells) in the middle was clearly observed (Fig. 2C). Additionally, several oval TM yeast cells had positive expression of CD86 and were highlighted in bright fuchsia (Fig. 2D). Interestingly, numerous bright red stripy areas of varying sizes were observed in the dermis at full view, indicating the regions of CD86-expressing macrophage aggregation. Upon closer inspection at high magnification, these areas revealed dense clusters or scattered TM yeasts inside and outside of macrophages (Fig. 3A). There were no similar obvious reminder signals in PAS staining (Fig. 3B). Obviously, CD86 immunohistochemical staining not only offered the advantage of easily identifying TM yeast cells in specimens but also demonstrated superior sensitivity compared to PAS staining in HIV-negative specimens (90.5% vs. 16.7%, $P <$ 0.05).

## Histopathologic characteristics of *Talaromyces marneffei* in HIV-positive biopsy specimens

The infiltration of inflammatory cells predominantly involved diffuse infiltration across different layers of the dermis, with most cases showing more severe inflammation in HIV-positive patients compared to HIV-negative cases infected with TM. In 10 HIV-positive specimens (71.4%), a large number of basophilic granular matter or round spore structures were found in H&E-stained specimens, primarily concentrated inside and around the dermal-infiltrated macrophages (https://github.com/fangjinling/Legend-of-Fig.-S1). Following PAS staining of all specimens, the results illustrated that 13

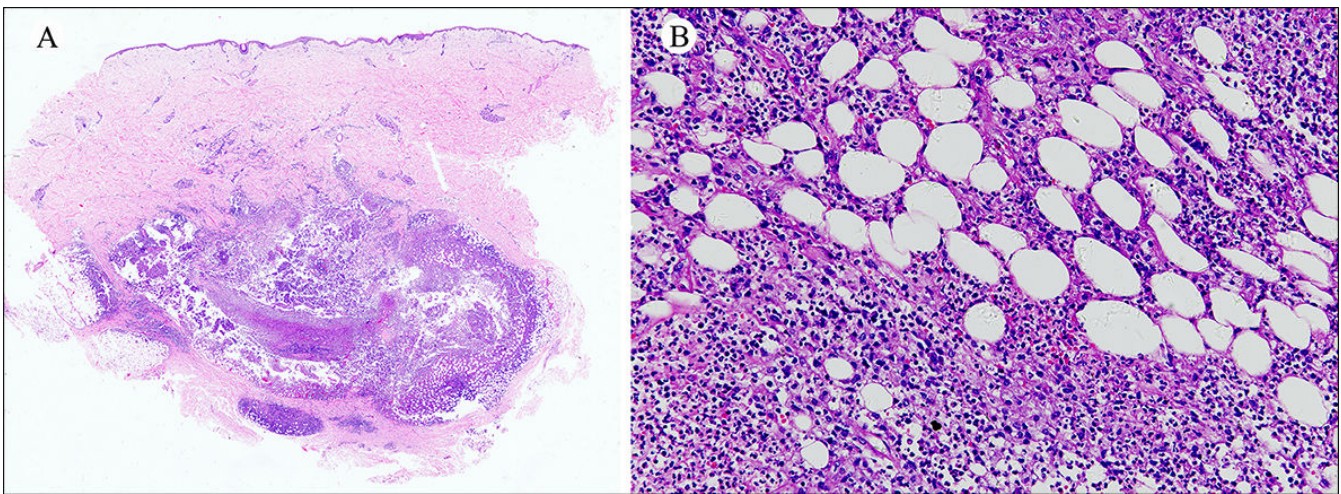

**FIG 1** Hematoxylin- and eosin-stained sections from an HIV-negative case of TM-infected lesion. (A) A full view of the infiltration of inflammatory cells into a dermal lesion. (B) Various inflammatory cell infiltrations, including histiocytes, lymphocytes, and multinucleated giant cells, were observed in the dermis. (A) Original magnification ×4. (B) Original magnification ×20.

HIV-positive sections (92.9%) were PAS-stained positive, including positive H&E-stained specimens. Mounds of TM yeast cells were infiltrated into the skin tissue, appearing as fuchsia spheres with empty halos and some clustered, resembling grapes (Fig. 4A). Under CD86 immunohistochemical staining, 14 HIV-positive sections (100%) were positive and showed a typical sausage-like TM yeast structure. Some intercellular yeasts formed

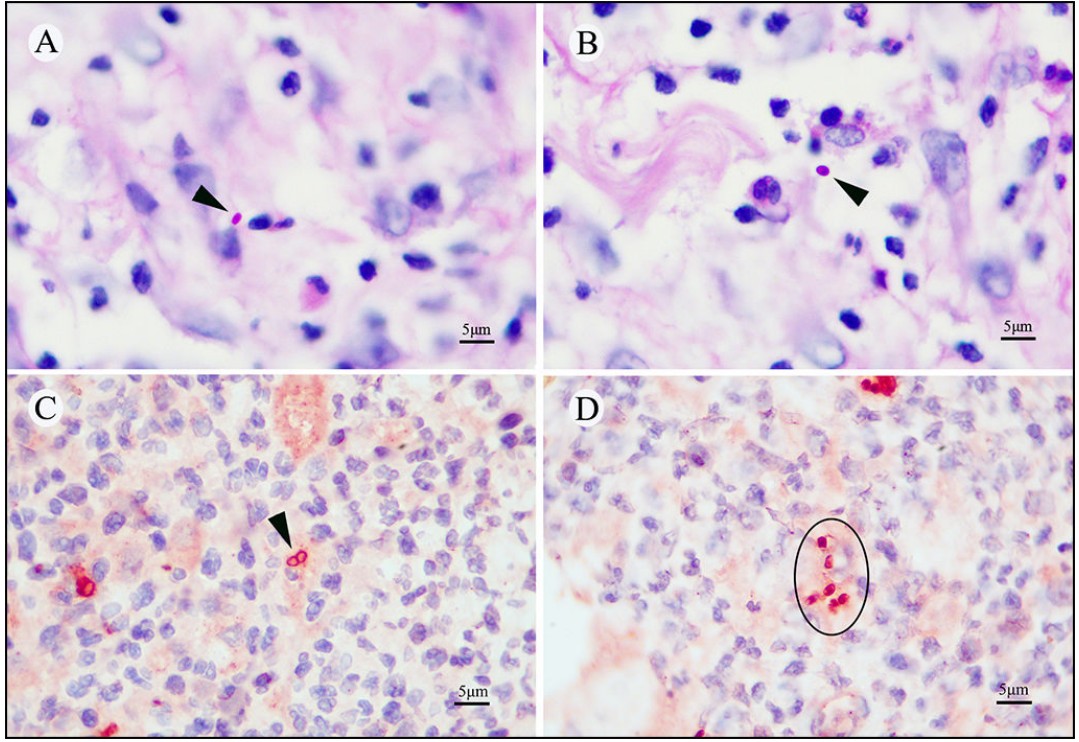

**FIG 2** CD86 immunohistochemical stained and periodic acid Schiff-stained sections from an HIV-negative case of TM-infected lesion. (A) In PAS staining, the long oval TM yeast cell appeared fuchsia without a deeply stained transverse septum (arrow). (B) The oval TM yeast cell was stained with fuchsia in PAS staining (arrow). (C) Macrophage cells and TM organisms had expression of CD86. The typical sausage-like yeast cell of TM, of which the cell wall and transverse septum were continuously stained with CD86 immunohistochemical staining (arrow). (D) Scattered TM yeast cells were observed in CD86-positive macrophage areas with darkly stained cell walls during CD86 immunohistochemical staining (circle). (A, B, C, D) Original magnification ×100.

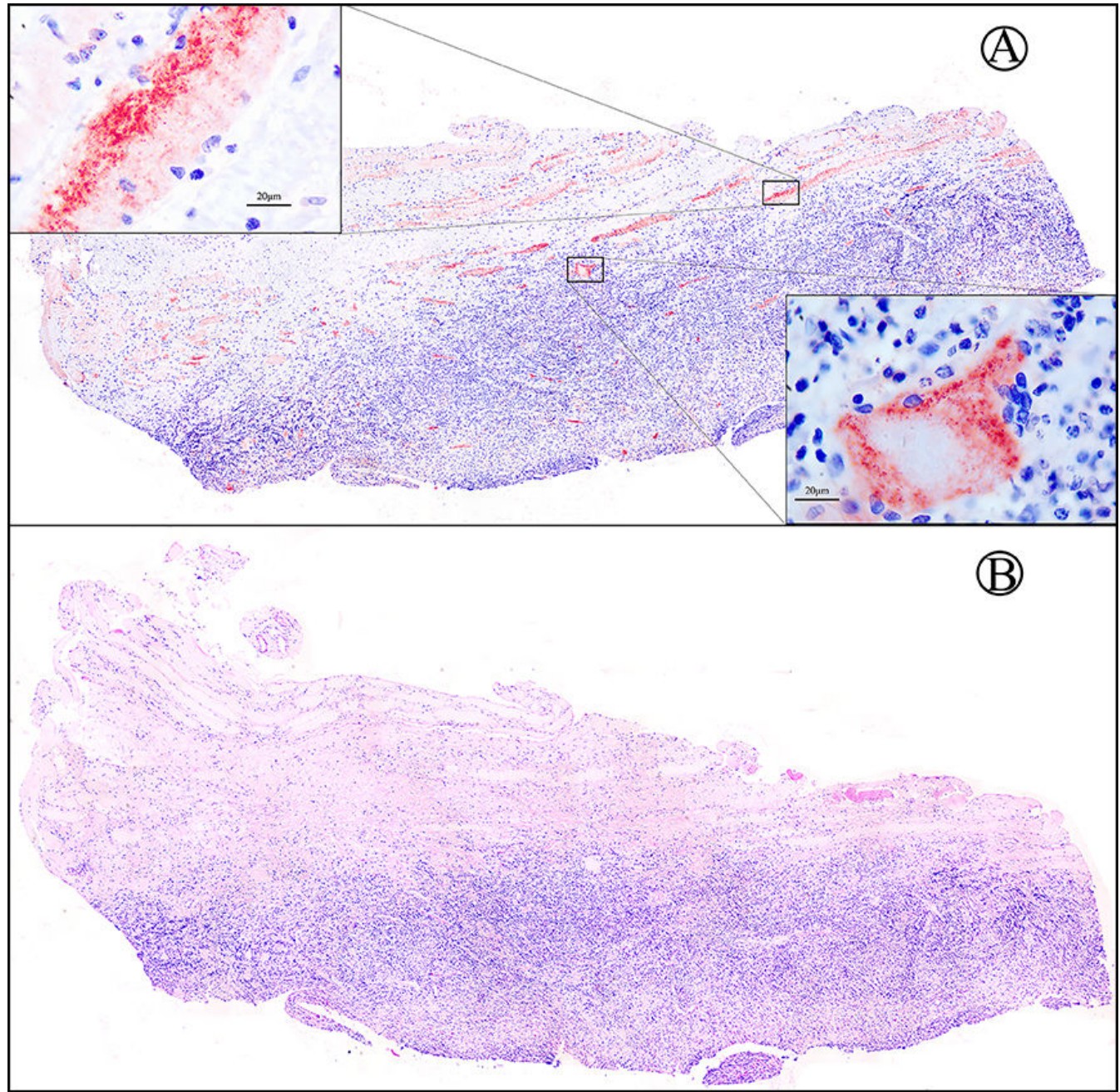

**FIG 3** The sections from a representative HIV-negative case of TM-infected lesion. (A) This image provides a comprehensive view of the skin lesions under CD86 immunohistochemical staining. There are a number of strip- or spot-like, red-stained areas in the dermis. Upon closer inspection with a high-power microscope, it is evident that these areas harbor a significant number of CD86-positive macrophages and TM yeast cells. (B) Conversely, the PAS staining of the same section did not exhibit such prominent signals. (A, B) Original magnification ×4.

clusters biased to one side, resembling signet ring cells; moreover, scattered yeasts were distinctly visible, appearing as small imprinted cells or bright red balls with hollow halos when observed microscopically (Fig. 4B). While CD86 immunohistochemical staining significantly outperformed PAS staining in terms of localizing and highlighting positive TM yeasts, there was no statistically significant difference in sensitivity between the two methods in HIV-positive specimens (100% vs. 92.9%, $P > 0.05$).

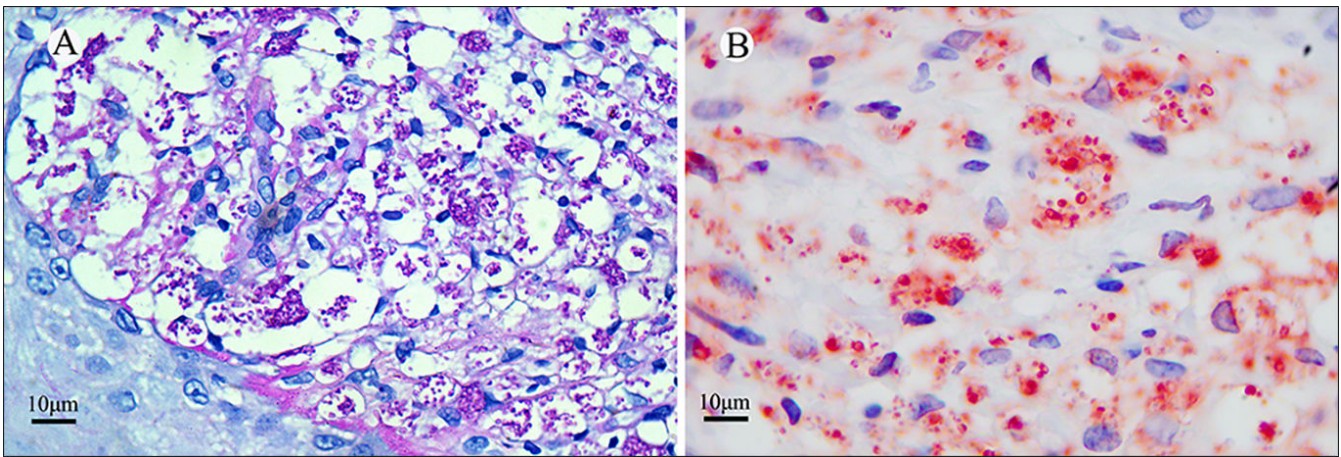

**FIG 4** CD86 immunohistochemical stained and periodic acid Schiff-stained sections from an HIV-positive case of TM-infected lesion. (A) Mounds of TM yeast cells were infiltrated into the skin tissue, appearing as fuchsia spheres with empty halos and some clustered, resembling grapes. (B) Many positive bright red TM organisms were seen within and outside the positive CD86-positive macrophages either in groups or scattered. (A) PAS and (B) CD86 stains. (A, B) Original magnification ×100.

## CD86 immunohistochemical staining of specimens from other fungal infections

In the study, 26 biopsy specimens of patients infected with various fungi were included, containing *Candida* (4 cases), *Cryptococcus* (4 cases), dematiaceous fungi (4 cases), *Histoplasma capsulatum* (1 case), *Aspergillus* (3 cases), *Sporothrix* (5 cases), *Trichosporon beigelii* (2 cases), *Mucor* (2 cases), and *Trichoderma*, and *Fusarium* spp. (1 case). The results indicated that macrophages expressing CD86 in skin lesions exhibited normal staining, whereas these fungal organisms did not show any staining. As shown in the figures of the CD86 staining, only CD86-positive macrophages were detected and stained red in the sections infected with *H. capsulatum*, whereas no CD86-positive staining was observed in the inclusion body of *H. capsulatum* (Fig. 5A). Similarly, only macrophages exhibited staining in the sections of lesions infected with *Cryptococcus*, with *Cryptococcus* showing no staining (Fig. 5B).

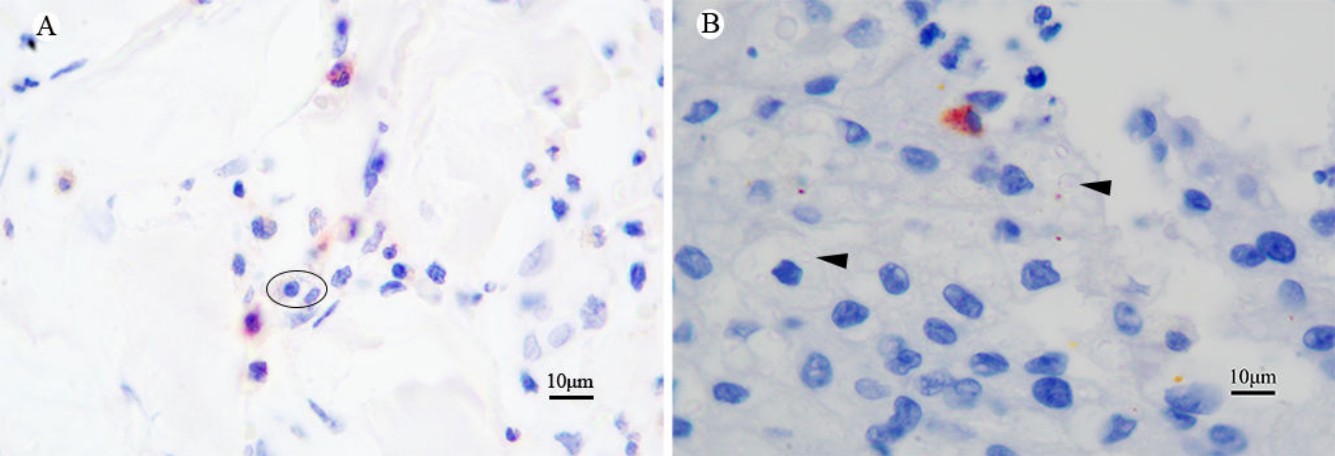

**FIG 5** CD86 immunohistochemical stained sections from cases of skin infected with *H. capsulatum* and *Cryptococcus*. (A) The section infected with *H. capsulatum* demonstrated CD86 protein expression in macrophages, with staining observed; however, the inclusion body-like *H. capsulatum* organisms inside macrophages were negative (circle). Additionally, (B) the section infected with *Cryptococcus* shows scattered, round, transparent Cryptococcus within the tissue, devoid of CD86-positive staining (arrow). (A, B) Original magnification ×100.

## DISCUSSION

This study presents a novel pathological diagnostic approach, CD86 immunohistochemical staining, to enhance the early detection of TM-infected lesions.Our findings demonstrated that CD86 immunohistochemistry can label not only macrophages but also TM organisms; the cell wall staining of positive TM was continuous and bright, making them easier to differentiate compared to PAS-stained yeasts. What makes it outstanding is the unique pathological features in CD86 immunohistochemical staining of TM-infected specimens: (i) at low magnification, the macrophage infiltration area in the dermis displays a ribbon-like or irregular red staining pattern; (ii) at high magnification, numerous red-stained TM yeast cells are visible in the intracellular and extracellular areas of the CD86-positive macrophages; and (iii) the cell wall and transverse septum of the characteristic sausage-like TM yeast cells display continuous bright red staining, which has a diagnostic value.

Furthermore, our data showed that CD86 immunohistochemistry can label not only macrophages but also TM organisms. For 26 non-TM tissue samples, the results indicated that CD86 staining specifically targeted macrophages in these tissue sections, with other fungi showing no staining. Statistically, the sensitivity and specificity were 92.9% and 100%, respectively, and CD86 immunohistochemical staining significantly outperformed PAS staining in terms of localizing and highlighting positive TM yeasts. It is worth noting that the sensitivity of CD86 staining was significantly higher than that of PAS staining in HIV-negative specimens (90.5% vs. 16.7%, $P < 0.05$) and showed comparable effectiveness in HIV-positive specimens (100% vs. 92.9%, $P > 0.05$). Owing to the intracellular fungal properties of TM and the low fungal loads in HIV-negative specimens, the positive rate of PAS staining was significantly lower than in HIV-positive specimens. In contrast, our previous studies have demonstrated that both intracellular and extracellular TM can capture CD86 molecules, resulting in both macrophages and a small amount of TM being positive in CD86 immunohistochemical staining. Therefore, there is not much difference for the CD86 immunohistochemical staining method.

Additionally, the absence of specific clinical manifestations of talaromycosis poses a significant challenge for clinicians in achieving early diagnosis. The clinical manifestations of TM-infected people are various and have individual differences between patients living with or without HIV (12, 20). In HIV-positive patients, the clinical manifestations are mainly fever, weight loss, weakness, anemia, and peculiar skin lesions, while in HIV-negative patients, they are commonly present with intermittent fever, widespread lymphadenopathy, and dermal lesions (1). Therefore, urgent efforts are needed to leverage the unique characteristics of TM for developing novel detection methods and maximizing the diagnostic benefits of pathological morphology to facilitate early diagnosis.

Rapid pathological diagnosis is based on the unique morphological features of TM observed under the microscope. Morphologically, TM exhibits the unique characteristic of thermal dimorphism, forming hyphae at 25°C and transitioning to a yeast phase at 37°C (21). TM yeast cells rely on fission, developing a transverse septum between cellular centers in the mitotic phase, resulting in the formation of typical sausage-like yeast cells (22). However, there are some fungi that are easily confused with TM in pathologic tissue. For instance, *H. capsulatum*, which shares a similar size with TM, is frequently observed in the form of inclusion bodies within infected tissues. Furthermore, certain *H. capsulatum* yeasts exhibit a narrow-necked morphology closely resembling the shape of sausage-like yeast cells. Additionally, the morphology of *Mucor* bears a resemblance to that of TM, which is characterized by a lightly stained region in the center dividing the cytoplasm into two halves. As *H. capsulatum* and *Mucor* base their reproduction on spores, they lack a transverse septum. The lightly or discontinuously stained cell wall and the unstained transverse septum of TM can lead to confusion between *Mucor*, *H. capsulatum*, and TM during PAS staining (23, 24). However, in CD86 immunohistochemical staining, the specimens of *Mucor* and *H. capsulatum* results indicated that no specific staining was observed for these fungi.

Current reports on pathological methods suggest that the round or oval shapes of TM yeasts with a cross-wall between the inside and outside of macrophages can be revealed by Grocott's methenamine silver or PAS staining. Wright's staining is more appropriate for cytological smears; the experimental specimens frequently include blood specimens, bone marrow specimens, and crushed tissue debris, which complicates the procedure for skin lesions. The similar-looking and -sized pathogens in bone marrow smears are difficult to distinguish in a single Wright's stain. Moreover, they do not specifically recognize TM yeast from their staining principle (7, 23, 25).

In conclusion, the CD86 immunohistochemical staining method plays an important role in the rapid pathological detection of cutaneous lesions, particularly in HIV-negative patients. We recommend utilizing the CD86 immunohistochemical staining in conjunction with PAS staining for the simultaneous detection of suspected TM infections to improve the early diagnosis rate of patients with TM.

## ACKNOWLEDGMENTS

The authors would like to acknowledge Shibin Luo and Xiaozhi Chen for immunohistochemical assistance.

We acknowledge funding support from the National Science Foundation of China (grant number 82260623) and the Natural Science Foundation of Guangxi Province (grant number 2022GXNSFAA035457).

D.L. contributed to the conceptualization of the experimental approach. J.F. and H.C. played a key role in data compilation and analysis and in performing the experiments. H.C. contributed to data collection and interpretation. J.F. prepared the first draft of the manuscript. K.H. plays an important role in grammar correction and reference checking. All authors reviewed the manuscript and provided comments. All authors were ultimately responsible for the decision to submit for publication.

## AUTHOR AFFILIATION

[1]Department of Dermatology, The First Affiliated Hospital of Guangxi Medical University, Nanning, China

## AUTHOR ORCIDs

Jinling Fang  http://orcid.org/0000-0002-8775-5208
Donghua Liu  http://orcid.org/0000-0002-5296-7665

## FUNDING

| Funder | Grant(s) | Author(s) |
| --- | --- | --- |
| National Science Foundation | 82260623 | Donghua Liu |
| Natural Science Foundation of Guangxi Zhuang Autonomous Region | 2022GXNSFAA035457 | Donghua Liu |

## AUTHOR CONTRIBUTIONS

Jinling Fang, Conceptualization, Data curation, Formal analysis, Investigation, Methodology, Software, Writing – original draft | Huiyuan Chen, Conceptualization, Data curation, Formal analysis, Investigation, Methodology, Software | Krishna Hamal, Writing – review and editing | Donghua Liu, Conceptualization, Funding acquisition, Methodology, Project administration, Resources, Supervision, Validation

## DATA AVAILABILITY

The data that support the findings of this study are available from the corresponding author upon reasonable request.

## ADDITIONAL FILES

The following material is available online.

### Open Peer Review

**PEER REVIEW HISTORY (review-history.pdf).** An accounting of the reviewer comments and feedback.

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
