## [Reviewer comments · Microbiology Spectrum]

Microbiology Spectrum

CD86 immunohistochemical staining for the detection of *Talaromyces marneffe* in lesions

Jinling Fang, Huiyuan Chen, Krishna Hamal, and Donghua Liu

Corresponding Author(s): Donghua Liu, The First Affiliated Hospital of Guangxi Medical University Department of Dermatology and Venereology

Review Timeline:

Submission Date:	August 20, 2024
Editorial Decision:	October 23, 2024
Revision Received:	November 18, 2024
Editorial Decision:	December 13, 2024
Revision Received:	December 19, 2024
Accepted:	January 13, 2025

Editor: Sadjia Bekal

Reviewer(s): The reviewers have opted to remain anonymous.

Transaction Report:

DOI: <https://doi.org/10.1128/spectrum.02063-24>

Re: Spectrum02063-24 (CD86 immunohistochemical staining for the detection of Talaromyces marneffeii in lesions)

Dear Dr. Donghua Liu:

Thank you for the privilege of reviewing your work. Below you will find my comments, instructions from the Spectrum editorial office, and the reviewer comments.

Revision Guidelines

Sincerely,
Sadjia Bekal
Editor
Microbiology Spectrum

Reviewer #1 (Comments for the Author):

The manuscript by J Fang et al. presents an interesting study developing a novel rapid diagnostic method for the important, emerging fungal infection talaromycosis, based on immunohistochemical staining. It is in general nicely written and the data supported the conclusion drawn. Below are some comments for the authors' consideration:

1. Scientific names should be written in the correct format. Please check throughout the manuscript.
2. Lines 34, 78-97 and throughout the manuscript: It appears to me that the CD86 molecule is a human host protein; yet based on previous studies *T. marneffeii* yeast cells are able to capture this CD86 molecule from macrophages so that they can also be stained with labelled anti-CD86 molecule antibodies. If this is true, then the authors should make clarification and explain this more clearly. The current writing is confusing since the sentences now read like *T. marneffeii* can produce and express the CD86 molecule themselves. If *T. marneffeii* does express CD86 molecule, that should be CD86 molecule-homologue?
3. Line 49: penicilliosis (small letter p)
4. Lines 60-61: If talaromycosis is ranked after tuberculosis, cryptococcosis and pneumocystis, then it should be the 4th most common HIV-associated opportunistic infection instead?
5. Lines 78-81: It is suggested that gene names, gene symbols and protein names follow the approved human gene nomenclature (<https://www.genenames.org/>). For example, CD86 molecule (not CD86 membrane protein), CD80 molecule, CD28 molecule and CTLA4 (not CTLA-4), etc.
6. Line 105: 'Dematiaceous fungi' is not a scientific name and need not be italicised.
7. Line 123: Do you mean 'Biopsies of the control group were solely incubated with antibody diluent and lacked anti-CD86 molecule antibodies'?
8. Lines 162, 169 & 170: Figure panels should be cited following their order of appearance in text. Please rearrange the panels of Figure 2.
9. Lines 191 & 196: Figure panels should be cited following their order of appearance in text. Please rearrange the panels of Figure 4.
10. Lines 202: The biopsy specimens are not of the various fungi. They came from the human patients. Please rewrite.
11. Lines 217 & 227: It may be inaccurate to state CD86 immunohistochemistry can specifically label TM organisms. The anti-CD86 molecule antibodies also stain macrophages or other APCs. Please rewrite to make the sentences clearer.
12. Lines 260-265: If the morphology of *Mucor* resembles that of TM in histopathology, any false positive of this CD86 immunohistochemistry on *Mucor*-positive specimens should be investigated as well.
13. Lines 268: Since Grocott's methenamine silver and Wright's staining can also be used for pathological diagnosis of talaromycosis, they should also be included as comparators when evaluating the performance of this CD86 immunohistochemistry.
14. Figure 2: For easier comparison, panels A & B as well as panels C & D should be shown in the same scale/magnification.
15. Line 410: TM bacteria?

Reviewer #2 (Comments for the Author):

The authors provide the CD86 immunohistochemical staining method that plays an important role in the rapid pathological detection of cutaneous lesions. The work is valuable in early diagnosis of TM infection, especially in HIV-negative patients. But can authors discuss the reason why the PAS staining yielded much higher sensitivity in HIV-positive specimens (13/14) than in the HIV-negative specimens (7/42) in the Discuss section, but there is no much difference for the CD86 immunohistochemical staining method. Is it due to the low fungal loads in cutaneous lesion tissue in HIV-negative patients than in the HIV-positive cases for the PAS staining? Other minor comments see below:

1. '*talaromyces marneffeii*' should be '*Talaromyces marneffeii*'
2. Line 158-178: Change the figure order of A-D in fig.2, as the description order from A to D in the text. Complemented the HE staining figures for HIV-negative specimens.
3. Line 177-178: 'superior sensitivity compared to PAS staining in HIV-negative specimens (90.5% vs.16.7%, $P < 0.05$)'. it is better to provide supplement figures of these PAS staining and CD86 immunohistochemical staining figures for the results supporting.
4. Line 183-184: 'with most cases showing more severe inflammation compared to HIV-negative cases infected with TM'. Clarify the description. 'with most cases showing more severe inflammation in HIV-positive patients compared to HIV-negative cases infected with TM'
5. Change the figure order of A and B in fig.4. complemented the HE staining figures for HIV-positive specimens.

The authors provide the CD86 immunohistochemical staining method that plays an important role in the rapid pathological detection of cutaneous lesions. The work is valuable in early diagnosis of TM infection, especially in HIV-negative patients. But can authors discuss the reason why the PAS staining yielded much higher sensitivity in HIV-positive specimens (13/14) than in the HIV-negative specimens (7/42) in the Discuss section, but there is no much difference for the CD86 immunohistochemical staining method. Is it due to the low fungal loads in cutaneous lesion tissue in HIV-negative patients than in the HIV-positive cases for the PAS staining? Other minor comments see below:

1. 'talaromyces marneffeii' should be '*Talaromyces marneffeii*'
2. Line 158-178: Change the figure order of A-D in fig.2, as the description order from A to D in the text. Complemented the HE staining figures for HIV-negative specimens.
3. Line 177-178: 'superior sensitivity compared to PAS staining in HIV-negative specimens (90.5% vs.16.7%, $P < 0.05$).'. it is better to provide supplement figures of these PAS staining and CD86 immunohistochemical staining figures for the results supporting.
4. Line 183-184: 'with most cases showing more severe inflammation compared to HIV-negative cases infected with TM'. Clarify the description. 'with most cases showing more severe inflammation in HIV-positive patients compared to HIV-negative cases infected with TM'
5. Change the figure order of A and B in fig.4. complemented the HE staining figures for HIV-positive specimens.

Dear Editors and Reviewers:

Thank you for giving us the opportunity to submit a revised draft of the manuscript “CD86 immunohistochemical staining for the detection of *Talaromyces marneffe* in lesions”. We appreciate the time and effort that you and the reviewers dedicated to providing feedback on our manuscript and are grateful for the insightful comments and valuable improvements on our paper. We have incorporated the suggestions made by the reviewers. Please reconsider our revised manuscript for publication.

Reviewer #1 (Comments for the Author):

The manuscript by J Fang et al. presents an interesting study developing a novel rapid diagnostic method for the important, emerging fungal infection talaromycosis, based on immunohistochemical staining. It is in general nicely written and the data supported the conclusion drawn. Below are some comments for the authors' consideration.

1. Scientific names should be written in the correct format. Please check throughout the manuscript. Lines 34, 78-97 and throughout the manuscript: It appears to me that the CD86 molecule is a human host protein; yet based on previous studies *T. marneffe* yeast cells are able to capture this CD86 molecule from macrophages so that they can also be stained with labelled anti-CD86 molecule antibodies. If this is true, then the authors should make clarification and explain this more clearly. The current writing is confusing since the sentences now read like *T. marneffe* can produce and express the CD86 molecule themselves. If *T. marneffe* does express CD86 molecule, that should be CD86 molecule-homologue?

Author response: Thank you for sharing your opinion. We agree with your opinion and revise the ‘We found that expression of CD86 was observed in TM pathogen’ as ‘We found that anti-CD86 antibody can label TM pathogen’ (Page 2 line 33-34).

2. Line 49: penicilliosis (small letter p)

Author response: Thank you for pointing this out. We have revised the ‘Penicilliosis’ to ‘penicilliosis’ (Page 3 line 48).

3. Lines 60-61: If talaromycosis is ranked after tuberculosis, cryptococcosis and pneumocystis,

then it should be the 4th most common HIV-associated opportunistic infection instead?

Author response: Thank you for pointing this out. The descriptions of original references:

‘The intersection with HIV has transformed *T. marneffe* from a rare human pathogen to a major cause of HIV-associated death, second only to tuberculosis and cryptococcosis or pneumocystis pneumonia in Thailand and Hong Kong.’

It can be seen that TM should be the 3rd most common HIV-associated opportunistic infection in Thailand and Hong Kong. We have revised the description as ‘Talaromycosis predominantly affects HIV-positive patients, with 16% prevalence in China, and is ranked after tuberculosis, and cryptococcosis or pneumocystis as the 3rd most common HIV-associated opportunistic infection in Thailand and Hong Kong.’ (Page 3 line 58-61).

4. Lines 78-81: It is suggested that gene names, gene symbols and protein names follow the approved human gene nomenclature (<https://www.genenames.org/>). For example, CD86 molecule (not CD86 membrane protein), CD80 molecule, CD28 molecule and CTLA4 (not CTLA-4), etc.

Author response: Thank you for your suggestions. We agree with your opinion and revise the ‘CD86 membrane protein’ ‘CTLA-4’ as ‘CD86 molecule’ ‘CTLA4’ (Page 4 line 76-84).

5. Line 105: ‘Dematiaceous fungi’ is not a scientific name and need not be italicised.

Author response: Thank you for pointing this out. We have corrected the errors in English usage (Page 5 line 103 and Page 10 line 201).

6. Line 123: Do you mean ‘Biopsies of the control group were solely incubated with antibody diluent and lacked anti-CD86 molecule antibodies’?

Author response: Yes, we have revised ‘Biopsies of the control group were solely incubated with antibody diluent and lacked anti-CD86 molecule antibodies’ to ‘Biopsies of the blank control group were solely incubated with antibody diluent and lacked anti-CD86 molecule antibodies; the remaining procedures were consistent’ (Page 6 line 120-122).

7. Lines 162, 169 & 170: Figure panels should be cited following their order of appearance in text. Please rearrange the panels of Figure 2.

Author response: Thank you for your suggestions. We have seriously revised the panels of figure2 and figure legend section of our manuscript as your advice (Page 20 line 391 - 400).

8. Lines 191 & 196: Figure panels should be cited following their order of appearance in text. Please rearrange the panels of Figure 4.

Author response: Thank you for your suggestions. We have seriously revised the panels of figure4 and figure legend section of our manuscript as your advice (from Page 20 line 410 to Page 21 line 415).

9. Lines 202: The biopsy specimens are not of the various fungi. They came from the human patients. Please rewrite.

Author response: Thank you for pointing this out. We agree with your opinion and revise the ‘26 biopsy specimens of various fungi’ to ‘26 biopsy specimens of patients infected with various fungi’ (Page 10 line 200).

10. Lines 217 & 227: It may be inaccurate to state CD86 immunohistochemistry can specifically label TM organisms. The anti-CD86 molecule antibodies also stain macrophages or other APCs. Please rewrite to make the sentences clearer.

Author response: Thank you for your suggestions. We agree with your opinion and revise the ‘CD86 immunohistochemistry can specifically label TM organisms’ as ‘CD86 immunohistochemistry can label not only macrophages but also TM organisms’ (Page 10 line 215-216 and Page 11 line 225-226).

11. Lines 260-265: If the morphology of *Mucor* resembles that of TM in histopathology, any false positive of this CD86 immunohistochemistry on *Mucor*-positive specimens should be investigated as well.

Author response: Thank you for sharing your opinion. *Mucor* and TM demonstrate similarity solely in specific sections; they can frequently be identified via H&E staining, characterized by broad-banded *Mucor* hyphae with bifurcation. Moreover, there were two cases in our specimens from patients with *Mucor* infection, neither of which was found anti-CD86 stain on *Mucor*. We

have complemented this description to the discussion. ‘However, in CD86 immunohistochemical staining, the specimens of *Mucor* and *H. capsulatum* indicated that no specific staining was observed for these fungi.’ (Page 13 line 265-267)

12. Lines 268: Since Grocott's methenamine silver and Wright's staining can also be used for pathological diagnosis of talaromycosis, they should also be included as comparators when evaluating the performance of this CD86 immunohistochemistry.

Author response: Thank you for your suggestions. On the one hand, Limited studies exist regarding the application of Grocott's methenamine silver (GMS) and Wright's staining in clinicopathologic diagnosis of talaromycosis, with most being case reports. Some studies indicate that the sensitivity of Wright staining is 52% and the sensitivity of GMS staining is 71%¹; the sensitivity of combined Wright and GMS staining is 72% in HIV-positive specimens². And many researchers have tested the application of PAS and GMS staining for the clinical diagnosis of onychomycosis indicate that PAS remains superior to GMS, while GMS demonstrates dominance only in a limited number of cases³.

One the other hand, the Wright's staining is more appropriate for cytological smears; the experimental specimens frequently include blood specimens, bone marrow specimens, and crushed tissue debris, which complicates the steps for skin lesions⁴. GMS staining is a cumbersome process that involves numerous unstable factors, posing a significant challenge for pathologists.

1. Supparatpinyo K, Chiewchanvit S, Hirunsri P, Uthammachai C, Nelson KE, Sirisanthana T. Penicillium marneffeii infection in patients infected with human immunodeficiency virus. Clin Infect Dis. 1992 Apr;14(4):871-4. doi: 10.1093/clinids/14.4.871. PMID: 1315586.

2. Ranjana KH, Priyokumar K, Singh TJ, Gupta ChC, Sharmila L, Singh PN, Chakrabarti A. Disseminated Penicillium marneffeii infection among HIV-infected patients in Manipur state, India. J Infect. 2002 Nov;45(4):268-71. doi: 10.1053/jinf.2002.1062. PMID: 12423616.

3. Shalin SC, Ferringer T, Cassarino DS. PAS and GMS utility in dermatopathology: Review of the current medical literature. J Cutan Pathol. 2020 Nov;47(11):1096-1102. doi: 10.1111/cup.13769. Epub 2020 Aug 22. PMID: 32515092.

4. Torlakovic E, Ames ED. Mycobacteria on Wright's-stained smears. Am J Clin Pathol. 1991 Aug;96(2):290. doi: 10.1093/ajcp/96.2.290. PMID: 1713743.

13. Figure 2: For easier comparison, panels A & B as well as panels C & D should be shown in the same scale/magnification.

Author response: Thank you for your suggestions. We have seriously unified the scale on panel of figure 2 as your advice.

14. Line 410: TM bacteria?

Author response: Thank you for pointing this out. We have revised the 'TM bacteria' as 'TM yeast cells' (Page 20 line 411).

Reviewer #2 (Comments for the Author):

The authors provide the CD86 immunohistochemical staining method that plays an important role in the rapid pathological detection of cutaneous lesions. The work is valuable in early diagnosis of TM infection, especially in HIV-negative patients.

1. But can authors discuss the reason why the PAS staining yielded much higher sensitivity in HIV-positive specimens (13/14) than in the HIV-negative specimens (7/42) in the Discuss section, but there is no much difference for the CD86 immunohistochemical staining method. Is it due to the low fungal loads in cutaneous lesion tissue in HIV-negative patients than in the HIV-positive cases for the PAS staining?

Author response: Thank you for sharing your opinion. 'Owing to the intracellular fungal properties of TM and the low fungal burden in HIV-negative specimens, the positive rate of PAS staining was significantly lower than in HIV-positive specimens. In contrast, our previous studies have demonstrated that both intracellular and extracellular TM can capture CD86 molecules, resulting in both macrophages and a small amount of TM were positive in CD86 immunohistochemical staining. Therefore, there is no much difference for the CD86 immunohistochemical staining method.' We have complemented this description to the discussion (Page 11 line 233-239).

Other minor comments see below:

2. '*talaromyces marneffeï*' should be '*Talaromyces marneffeï*'

Author response: Thank you for pointing this out. We agree with your opinion and revise the '*talaromyces marneffeï*' as '*Talaromyces marneffeï*'.

3. Line 158-178: Change the figure order of A-D in fig.2, as the description order from A to D in the text. Complemented the HE staining figures for HIV-negative specimens.

Author response: Thank you for your suggestions. We have seriously revised the panels of figure2 and figure legend section of our manuscript as your advice. And the Fig. 1 is the HE staining figures for HIV-negative specimens.

4. Line 177-178: 'superior sensitivity compared to PAS staining in HIV-negative specimens (90.5% vs.16.7%, $P < 0.05$)'. it is better to provide supplement figures of these PAS staining and CD86 immunohistochemical staining figures for the results supporting.

Author response: Thank you for your suggestions. The Fig.2 and Fig.3 both illustrate the difference between PAS staining and CD86 immunohistochemical staining in the same specimen.

5. Line 183-184: 'with most cases showing more severe inflammation compared to HIV-negative cases infected with TM'. Clarify the description. 'with most cases showing more severe inflammation in HIV-positive patients compared to HIV-negative cases infected with TM'

Author response: Thank you for your suggestions. We agree with your opinion and revise the 'with most cases showing more severe inflammation compared to HIV-negative cases infected with TM' as 'with most cases showing more severe inflammation in HIV-positive patients compared to HIV-negative cases infected with TM' (Page 9 line181-182).

6. Change the figure order of A and B in fig.4. complemented the HE staining figures for HIV-positive specimens.

Author response: Thank you for your suggestions. We have seriously revised the panels of figure4 and figure legend section of our manuscript as your advice and have complemented the

HE staining figures for HIV-positive specimens as supplemental material Fig. S1.

Sincerely,

Donghua Liu

Re: Spectrum02063-24R1 (CD86 immunohistochemical staining for the detection of *Talaromyces marneffe* in lesions)

Dear Dr. Donghua Liu:

Thank you for the privilege of reviewing your work. Below you will find my comments, instructions from the Spectrum editorial office, and the reviewer comments.

Revision Guidelines

Sincerely,
Sadjia Bekal
Editor
Microbiology Spectrum

Reviewer #1 (Comments for the Author):

The authors have largely addressed comments from the previous round of reviewers. However, the use of English in the manuscript still needs improvement. Many grammatical errors are spotted. Below are some further suggestions:

Line 99: The presentation here is still confusing. So TM yeasts do produce CD86 molecular-homologue? Please provide the

reference.

Line 102: What is 'the control group'?

Line 127: Please provide the ethics approval number.

Dear Editors and Reviewers:

Thank you for giving us the opportunity to submit a revised draft of the manuscript “CD86 immunohistochemical staining for the detection of *Talaromyces marneffe* in lesions”. We appreciate the time and effort that you and the reviewers dedicated to providing feedback on our manuscript and are grateful for the insightful comments and valuable improvements on our paper. We have incorporated the suggestions made by the reviewers. Please reconsider our revised manuscript for publication.

Reviewer #1 (Comments for the Author):

The authors have largely addressed comments from the previous round of reviewers. However, the use of English in the manuscript still needs improvement. Many grammatical errors are spotted.

Author response: We are honoured to receive your careful comments. The errors in usage and grammar have been corrected with the help of professionals.

Below are some further suggestions:

Line 99: The presentation here is still confusing. So TM yeasts do produce CD86 molecular-homologue? Please provide the reference.

Author response: Thank you for your patience and guidance! In our previous study, we constructed the co-culture model of THP-1 macrophages and *Talaromyces marneffe* (TM). We used confocal fluorescence microscopy, immunoelectron microscopy, indirect immunofluorescence tests, and immunohistochemistry assays to investigate the relationship between CD86 molecules and TM organisms. We demonstrated that TM yeasts can capture CD86 molecules from macrophages in vitro instead of producing CD86 molecular- homologue.¹

We have revised the ‘in animal experiments, the CD86 molecule-homologue of TM yeasts in lung and liver tissue sections of mice infected with TM was observed.’ as ‘we discovered that TM binds to CD86 monoclonal antibodies in the lung and liver tissues of mouse models infected with TM.’ (Page 5 line 96-98)

1. Fang J, Chen R, Liu D. *Talaromyces marneffeii* Can Capture CD86 Proteins of Macrophages in vitro. *Infect Drug Resist.* 2022 Nov 25;15:6801-6810. doi: 10.2147/IDR.S389612. PMID: 36458198; PMCID: PMC9707389.

Line 102: What is 'the control group'?

Author response: The control groups included the negative section control and no primary control. The no primary control sections were solely incubated with antibody diluent and lacked anti-CD86 molecule antibodies, the negative section control sections were without primary and secondary antibodies, and the remaining procedures were consistent.

We have complemented this description to the discussion.

'Eighty-two tissue samples were collected and sliced to a thickness of 2-3 μ m, and the slices from each sample were assigned to three groups: the experimental group, the negative section control, and no primary control.' (Page 6 line 119 - 121)

'The no primary control sections were solely incubated with antibody diluent and lacked anti-CD86 molecule antibodies, the negative section control sections were without primary and secondary antibodies, and the remaining procedures were consistent.' (from Page 6 line 132 to Page 7 line 135)

Line 127: Please provide the ethics approval number.

Author response: Thank you for pointing this out. We have supplemented the ethics approval number in the manuscript (Page 4 line 116).

Sincerely,

Donghua Liu

Re: Spectrum02063-24R2 (CD86 immunohistochemical staining for the detection of Talaromyces marneffeii in lesions)

Dear Dr. Donghua Liu:

Your manuscript has been accepted, and I am forwarding it to the ASM production staff for publication. Your paper will first be checked to make sure all elements meet the technical requirements. ASM staff will contact you if anything needs to be revised before copyediting and production can begin. Otherwise, you will be notified when your proofs are ready to be viewed.

Sincerely,
Sadjia Bekal
Editor
Microbiology Spectrum

Reviewer #1 (Comments for the Author):

The authors have addressed all comments from the previous round of review.